# A double ovulation protocol for Xenopus laevis produces doubled fertilisation yield and moderately transiently elevated corticosterone levels without loss of egg quality

Chloe Moss[1], Barbara Vacca[1], Jo Arnold[2], Chantal Hubens[1], Dominic M. Lynch[1], James Pegge[1], Michael A. R. Green[3], Charlotte A. Hosie[2], Tessa E. Smith[2], Jeremy B. A. Green[1]*

**1** Centre for Craniofacial Regeneration and Biology, King's College London, London, United Kingdom, **2** Department of Biological Sciences, University of Chester, Chester, United Kingdom, **3** Dept of Economics, University of Oxford, Oxford, United Kingdom

☯ These authors contributed equally to this work.

* jeremy.green@kcl.ac.uk

## Abstract

The African claw-toed frog, *Xenopus laevis*, is a well-established laboratory model for the biology of vertebrate oogenesis, fertilisation, and development at embryonic, larval, and metamorphic stages. For ovulation, *X. laevis* females are usually injected with chorionic gonadotropin, whereupon they lay typically hundreds to thousands of eggs in a day. After being rested for a minimum of three months, animals are re-used. The literature suggests that adult females can lay much larger numbers of eggs in a short period. Here, we compared the standard "single ovulation" protocol with a "double ovulation" protocol, in which females were ovulated, then re-ovulated after seven days and then rested for three months before re-use. We quantified egg number, fertilisation rate (development to cleavage stage), and corticosterone secretion rate as a measure of stress response for the two protocol groups over seven 3-month cycles. We found no differences in egg number-per-ovulation or egg quality between the groups and no long-term changes in any measures over the 21-month trial period. Corticosterone secretion was elevated by ovulation, similarly for the single ovulation as for the first ovulation in the double-ovulation protocol, but more highly for the second ovulation (to a level comparable to that seen following shipment) in the latter. However, both groups exhibited the same baseline secretion rates by the time of the subsequent cycle. Double ovulation is thus transiently more stressful/demanding than single ovulation but within the levels routinely experienced by laboratory *X. laevis*. Noting that "stress hormone" corticosterone/cortisol secretion is linked to physiological processes, such as ovulation, that are not necessarily harmful to the individual, we suggest that the benefits of a doubling in egg yield-per-cycle per animal without loss of egg quality or signs of acute or long-term harm may outweigh the relatively modest and transient corticosterone elevation we observed. The double ovulation protocol therefore represents a potential new standard

**Data Availability Statement:** All relevant data are within the manuscript and its Supporting Information files.

**Funding:** This work was funded by National Centre for the Replacement Refinement and Reduction of Animals in Research (NC3Rs) grant NC/S000933/1-GREEN to J.B.A.G. (PI), T.E.S. (Co-I) and C.A.H. (Co-I). The funders had no role in study design, data collection and analysis, decision to publish, or preparation of the manuscript.

**Competing interests:** The authors have declared that no competing interests exist.

practice for promoting the "3Rs" (animal use reduction, refinement and replacement) mission for Xenopus research.

## Introduction

*Xenopus laevis*, the African claw-toed frogs, is an essential vertebrate model system for biomedical research with a long history, ongoing major investment, and vigorous publication and citation output [1]. *Xenopus* has been instrumental in a wide range of biological discoveries, including recent breakthroughs in stem cell research [2–4], regenerative medicine [5, 6], human disease genetics, including for cancer [7, 8], Alzheimer's disease [9], heart disease [10, 11], kidney disease [12, 13], diabetes [14–16], and craniofacial and auditory malformations [17, 18]. *Xenopus* has many advantages as a model organism (including large eggs and embryos, large egg clutches (thousands per female per day), simple growth medium and non-dilution of lineage tracers due to reduction-division development. These advantages keep it as a pioneering laboratory animal. Improving methods for its use as a laboratory model while ensuring good welfare standards is therefore a priority. This is not only from an ethical but also a scientific standpoint, since we know that animals with good well-being serve as more valid, robust scientific models compared to animals with poorer well-being [19].

Data from an informal worldwide survey of 210 laboratories that use *Xenopus laevis* in developmental studies (C. James-Zorn & E. Pearl, Xenbase.org, unpublished data) suggest a current world population of laboratory *Xenopus* in this group of over 52,000 animals. 95% of these labs use hormone-induced ovulation to obtain eggs. Improving the ovulation protocol to increase egg quantity and quality while maintaining good welfare potentially offers considerable benefits in cost-savings and animal-use reduction.

The current standard *X. laevis* ovulation protocol allows females to be used repeatedly for many years with three-month rest periods between successive hormone-induced ovulations [20] (with previous standard works suggesting 2–3 months [21, 22]). The rationale for the three-month rest period is unclear. *X. laevis* oocytes take five to seven weeks to mature [23, 24], which means that during a three-month rest period an entirely fresh cohort of stage I oocytes can undergo complete maturation. However, since normal ovaries contain oocytes at all stages of maturation, there will be many additional maturing oocytes in a three-month period. This maturation turnover suggests that *X. laevis* can generate many mature eggs in a much shorter time. This is consistent with field reports that suggest that in the wild *X. laevis* females mate multiple times in a single rainy season [25]. Historically laboratory Xenopus were rested anywhere from a week or two [23, 24, 26, 27] to six months or longer [28, 29]. A *X. laevis* induced-ovulation conducted weekly for 16 weeks resulted in no reduction of egg numbers laid in the second week and only a 50% reduction to 1000 eggs/animal after four weeks [24]. These observations suggest that ovulating females twice over two weeks with three months rest in between ("double ovulation") could double productivity without depleting egg reserves, enabling twice as many eggs to be collected (and twice as many experimental days) per frog per calendar period or the same number of eggs to be collected from half the number of frogs. This latter scenario supports the "Reduction" component of the "3R's" (scientific animal use Reduction, Refinement, and Replacement) framework.

Pilot experiments in one of our labs (J.B.A.G.) produced no discernible adverse effects of double ovulation (i.e. no obvious acute health or productivity changes compared to single-ovulated animals). However, to establish whether double ovulation could be widely permissible as

a standard for Xenopus research, we set out to investigate whether its chronic use is sustainable in terms of egg quantity and quality and whether there would be no substantial harm to the animals. Ensuring healthy frogs that produce large numbers of high-quality eggs is important both ethically and because animals with a poor physical and mental states make poor scientific models in the laboratory due to a reduction in the reliability and repeatability of scientific results obtained from them [19].

Amphibians possess a hypothalamic-pituitary-inter-renal (HPI) axis (akin to the mammalian hypothalamic-pituitary-adrenal axis) which, in addition to controlling physiological processes such as energy mobilization [30], is activated under conditions of stress [31]. The activated HPI axis releases a group of hormones, including corticosterone which has a broadly equivalent role in amphibians to cortisol in other taxa, and triggers a cascade of physiological reactions to enable the animal to cope with the stressor and re-establish homeostasis. We have previously validated methods to quantify water-borne cortisol in *X. laevis* and shown these non-invasive methods reliably detect significant increases in this glucocorticoid in response to stress such as transport [32, 33].

Here we present the results of a 21 month trial of double ovulation in *X. laevis* as a potential standard for laboratory use of this species. In this trial, we monitored not only egg yield and fertilisability (a measure of egg quality relevant to the typical use in developmental biological studies) but also levels of corticosterone, the amphibian counterpart of corticosteroids associated with managing stress [32–34]. We found that double ovulation for the entire period caused no reduction in egg yield or quality, or general health of the experimental frogs compared to matched females experiencing single ovulation. In both single- and double-ovulation groups corticosterone levels were elevated acutely by the ovulation protocol, with a somewhat higher spike for the second ovulation in the latter. No chronic effects on baseline or peak levels of corticosterone secretion were observed.

## Results

### Double ovulation yields double the number of eggs per frog per cycle

Details of the trial design are given in the Methods section. In brief, we compared the egg yield and fertilisability of two groups of 40 weight-matched frogs in a randomised trial. One group was ovulated once per three-month cycle (as per existing standard practice) and the other group ovulated twice per cycle, with the second ovulation a week after the first. The staff performing the hCG injections, egg collections and all frog handling were blinded to which frog was in which group at all times. Unblinding was done only at trial completion. We collected data on the first two three-month cycles and last two cycles of seven cycles in total (21 months) to determine acute and chronic effects of the two ovulation protocols respectively. **Fig 1A, 1B, 1E and 1F** shows the resulting egg yields. Although there was considerable variability for all of the different groups and cycles, the general pattern was that frogs laid about as many eggs in the second ovulation of a cycle as on the first and single ovulations. Mean ± S.D. values were 3144 ± 2245, 2813 ± 2338, and 2890 ± 2370 for single, first-of-double, and second-of-double ovulations respectively. There were no statistically significant differences between the egg numbers for any of the conditions (single, first-of-double, second-of-double) either within any single cycle or when comparing conditions using pooled values (i.e. pooling the two early and two late cycles) (p > 0.05, Tukey tests following one-way ANOVA). Thus, per 3-month cycle, the double-ovulated frogs produced roughly twice as many eggs as the single-ovulated frogs. This was the case both at the beginning of the trial (cycles 1 and 2) and the end of the trial (cycles 6 and 7) indicating that there was no sign of egg depletion or exhaustion following repeated double ovulation every three months over a 21-month period. A very slightly lower

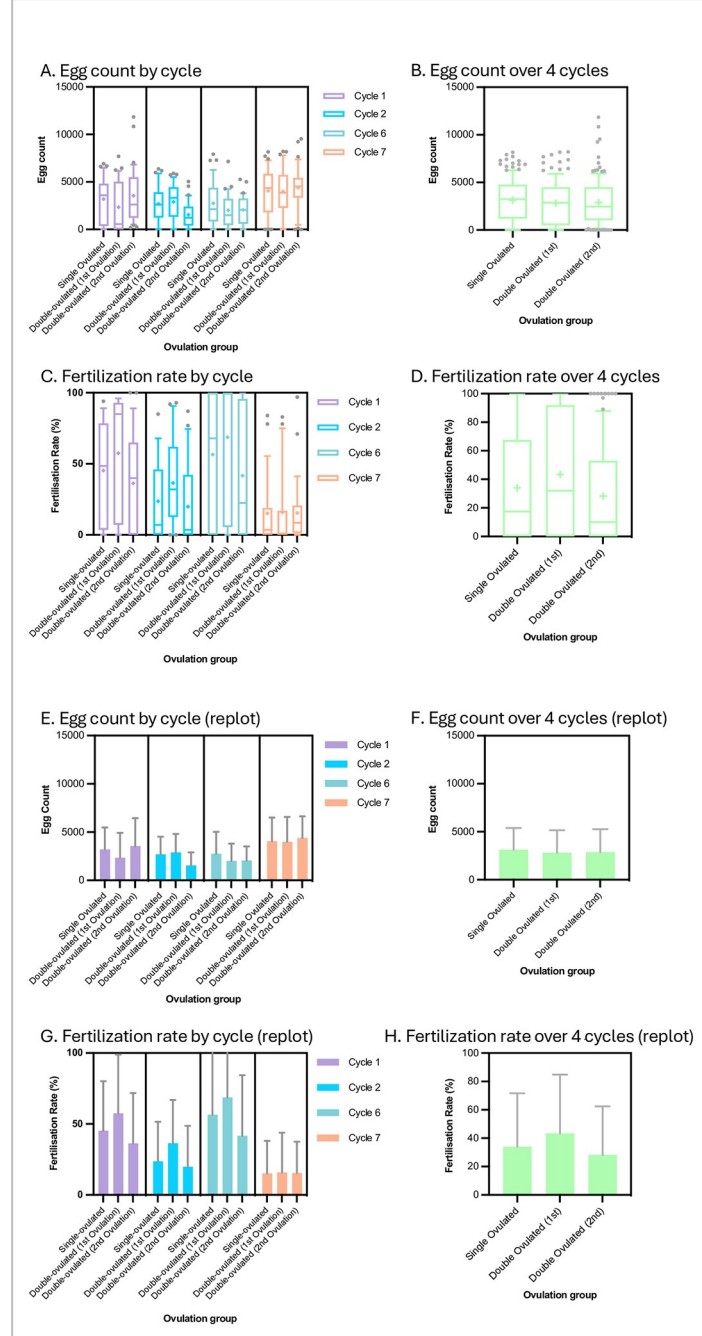

**Fig 1. *X. laevis* egg yields and fertilisation rates under single- and double-ovulation protocols are similar. A, E, B, F:** Numbers of eggs laid in 10 hours (9.00 am to 7.00 pm following hCG injection the previous evening) for each condition in the specific ovulation cycles shown (A, E) and for all four ovulation cycles pooled (B, F). **C, G, D, H:** Percentage of fertilised (cleaving) eggs from a ~200 egg sample in the specific ovulation cycles shown (C, G) and for all four ovulation cycles pooled (D, H). Data are plotted as box-and-whisker format (whiskers at 10/90th percentiles, bar at median, cross at mean) to show variability (A-D) and re-plotted as histograms (mean +/- standard deviation) (E-H) to show trends at a glance. Sample numbers: n ≥ 31 for each condition in each cycle and so ≥ 124 for pooled cycles (see Supporting Information S1 Appendix for raw data).

egg yield could be detected in the double-ovulated group (**Fig 1**), but this was not statistically significant (ANOVA and Tukey tests) and did not appear to indicate a long-term trend.

## Double ovulation produces slightly lower egg fertilisability in the second ovulation

Although not reported in the literature, egg quality is notoriously variable in Xenopus and high variability is apparent in the results plotted in **Fig 1C, 1D, 1G and 1H**. Mean ± S.D. values of percentage of fertilised eggs reaching cleavage stage were 34.05 ± 37.64%, 43.37 ± 41.4%, and 28.22 ± 34.21% for single, first-of-double, and second-of-double ovulations respectively. We were concerned that frogs producing more eggs might produce eggs of lower quality, but there were no statistically significant differences between fertilisation rates of eggs the single-ovulated and either ovulation of the double-ovulated group in any single cycle (p > 0.05, Tukey tests following one-way ANOVA). There was a slightly lower fertilisation rate for the second ovulation eggs in each cycle which was not statistically significant. This became a statistically significantly lower fertilisation rate in the second versus the first ovulation of the double-ovulation group when values for the four cycles were pooled (p = 0.01, Tukey test following one-way ANOVA). This may have been to do with the "starting" fertilisation rate (i.e. in the first ovulation of the first cycle) being slightly (albeit not statistically significantly) higher in the group due to be double-ovulated compared to the single-ovulation group. Since the different treatment groups were weight-matched, co-housed and had all been ovulated just once at this point, this was presumably background noise.

There was a substantial reduction in egg quality in the last (seventh) cycle in both the treatment and control groups (no statistically significant difference) for undetermined reasons.

## Corticosterone secretion levels are elevated by the ovulation protocol, slightly more so in the second of a double ovulation, but return to baseline within a week

Neither the egg yield and quality nor the general health and appearance of the frogs showed any obvious signs of difference between the single- and double-ovulated groups, suggesting no significant harm resulted from the higher demands on the latter. However, we measured corticosterone secretion to gauge the level of stress and overall physiological demands on each group that could have been "sub-clinical". To this end, we collected one-hour-conditioned water samples from individual frogs before and after each ovulation and assayed excreted corticosterone according to our previous protocol [32, see Methods for details].

We found that the baseline corticosterone secretion levels were highly variable. They averaged 1126 ± 800 pg/hr, which was similar to levels previously reported for *X. laevis* [32, 33] (**Table 1**). Means ± S.D.s for single-ovulated, double-ovulated-1[st]-ovulation and double-ovulated-2[nd]-ovulation groups were, respectively, 1134 ± 841, 1073 ± 687 and 1170 ± 732 pg/hr (compared with 1400 ± 600 in [32]).

Unsurprisingly, given both the physical handling and the physiological demands that hCG stimulation imposes, the ovulation procedure increased corticosterone secretion in both groups (**Fig 2A and 2B**). None of the increases was statistically significant in individual cycles, but, presumably due to the high variability, when cycles were pooled, increases became statistically significant (p < 0.05, Tukey tests following ANOVA). Secretion was similar between the single-ovulated and the first ovulation of the double-ovulated group, rising to 1591 ± 865 and 1652 ± 1124 pg/hr (mean ± S.D.) (p = 0.99, Tukey test) respectively. However, although the corticosterone secretion rate was back down to baseline after a week, the second ovulation of the double-ovulated group produced a corticosterone secretion rate increase after ovulation

**Table 1. Summary of female corticosterone secretion rates.**

| Condition | Corticosterone secretion |
|---|---|
| | Mean ± S.D. (pg/hr) |
| Single ovulated–pre ovulation | 1134 ± 841 |
| Single ovulated–post ovulation | 1591 ± 865 |
| Double ovulated–pre-1st-ovulation | 1073 ± 687 |
| Double ovulated–post-1st-ovulation | 1652 ± 1124 |
| Double ovulated–pre-2nd-ovulation | 1170 ± 732 |
| Double ovulated–post-2nd-ovulation | 2470 ± 1513 |
| Baseline–black tanks (from [33]) | 1000 ± 600 |
| Baseline–white tanks (from [33]) | 1350 ± 600 |
| Pre-shipping (from [32]) | 1400 ± 600 |
| Post-shipping (from [32]) | 2500 ± 600 |

(to an average of 2470 ± 1513 pg/hr) that was statistically significantly higher than that for both the single-ovulation group and first ovulation of the double-ovulation group ($p < 0.0001$ for both, Tukey test following one-way ANOVA). We further conducted statistical equivalence tests known as two-one-sided-t-tests (TOST tests) asking whether despite being statistically significantly higher the corticosterone levels were consistent with being equivalent (within 30%) in the second ovulation versus the single/first ovulation. This 30% criterion was based on the similar difference in levels between frogs in white versus black tanks [33], which we considered to be a change below which the levels could be considered equivalent. We found that even using this looser test, the corticosterone levels after the second of the two ovulations in the double-ovulation group was not equivalent to the level after the single ovulation ($p = 0.99$).

Sample numbers: $n \geq 31$ for each condition in each cycle and so $\geq 124$ for pooled cycles.

Across all tested conditions, there were no signs of chronic changes, All corticosterone secretion rates were not only back down to baseline by the following ovulation cycle, but also no more elevated by ovulation in the final (7th) cycle than in earlier cycles (no statistically significant differences in pre-ovulation values, $p > 0.05$, tukey test following one-way ANOVA).

## Body weights showed the same growth over the trial under single- and double-ovulation regimes

The most closely weight-matched pairs of frogs from each treatment group at the beginning of the trial, which were co-housed in the same tanks to control for tank effects, were re-weighed at the end of the trial. Weights increased on average by just over 1.5-fold in both groups (from 124 ± 10 g to 195 ± 24 g in the single-ovulation group and 125 ± 11 g to 192 ± 18 g in the double ovulation group (mean ± SD). Weight increases were not statistically significantly different between the two groups ($p = 0.53$, paired t-test) (**Fig 3**).

## Discussion

Our trial clearly showed that over the course of seven three-month cycles our double-ovulation protocol yielded approximately double the number of eggs as the traditional single ovulation protocol. Similar numbers of fertilisable eggs were obtained using the double-ovulation protocol on twice as many days as using single ovulation, and there was no overt difference between the protocols in acute or chronic welfare of the frogs. This result confirms the premise that the egg-laying capacity of this species is much higher than is used by standard practice. A recommendation to adopt a double ovulation protocol, to enable either the halving of colony size or

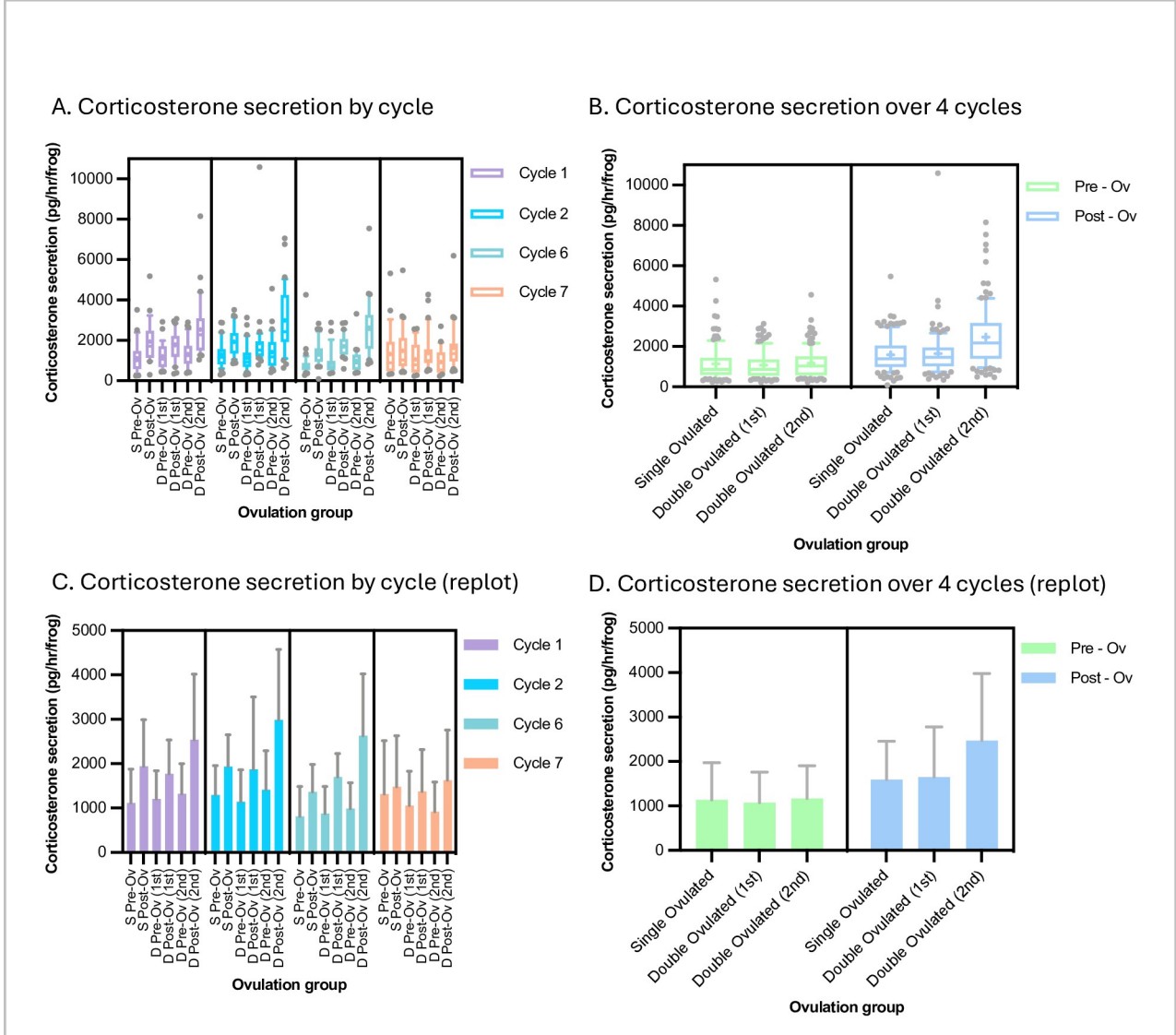

**Fig 2. *X. laevis* corticosterone secretion under single- and double-ovulation protocols.** Corticosterone secretion (pg/hr) values for each condition in the specific ovulation cycles shown (panels **A** and **C**) and for all four ovulation cycles pooled (panels **B** and **D**). Data are plotted as box-and-whisker format (whiskers at 10/90th percentiles, bar at median, cross at mean) to show variability (panels A and C) and re-plotted as histograms (mean +/- standard deviation) (panels B and D) to show trends at a glance.

doubling output would therefore seem reasonable and would fulfil the second of the "3Rs" (replacement, reduction, refinement) mission with respect to reduction of laboratory animal use.

Some potential limitations of this conclusion should, however, be considered. First, there was variability in the conditions in our animal facility and in both animal care staff and researchers on this project and this may have obscured small but statistically significant differences between the results for different regimes. In practice, however, the co-housing of the frogs in the two treatment groups in the same tanks in all cases, as well as careful blinding of the care staff and experimenters to the identities of the frogs during the trial controlled for these differences so that we can be confident that similarities and differences we measured are real effects of the different treatments.

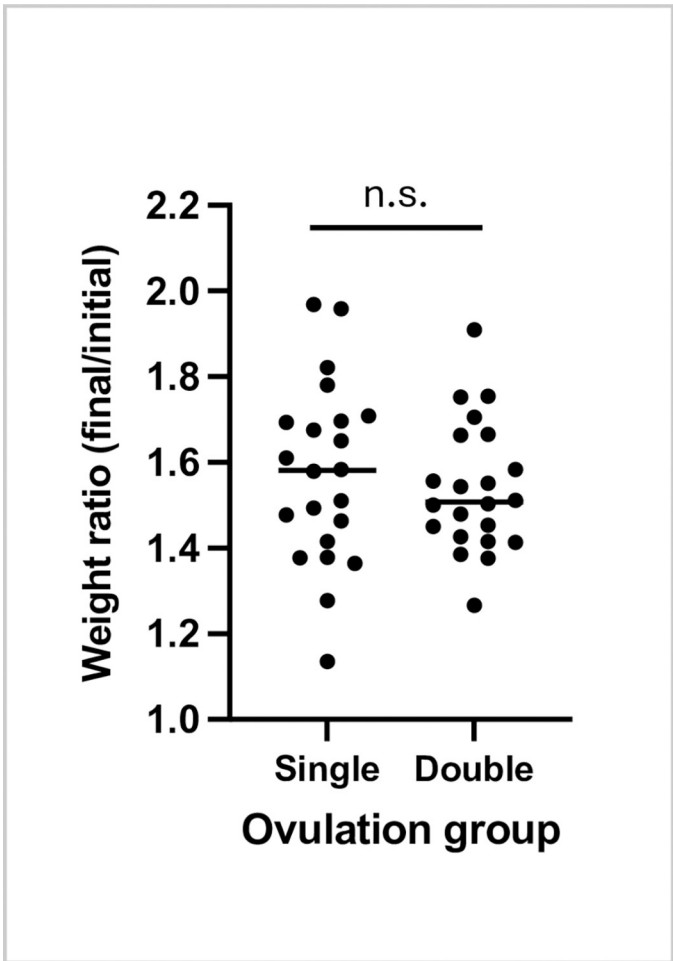

**Fig 3. *X. laevis* female weights increased similarly under single- and double-ovulation protocols.** Ratios of weights at the beginning and end of the trial (21 months) as plotted showing mean +/- standard deviation showing no significant difference between the groups (two-tailed paired t-test.

Second, it could be argued that some further measurements that could have been informative were not made. These included the time-course of the return-to-baseline of corticosterone levels following the second of the ovulations in the double ovulation regime, the weights and growth rate of the animals in each group in the middle of the trial, and, of course, effects beyond the period of the trial. It would indeed be interesting to track differences the recovery time for corticosterone levels after ovulation, but for this study we first wanted to establish whether there were any differences at all, and did not want to increase the amount of handling that would have been needed for additional corticosterone and weighing measurements (which could themselves add stress). Likewise, an even longer time course would be interesting, but the length of the trial was constrained by the resources available so that further chronic effects of double ovulation could not be determined. Nonetheless, following our study, each of these questions can now be addressed with an appropriately designed long-term follow-up.

Third, our results showed that although a second ovulation per cycle was sustainable, it did produce corticosterone secretion statistically significantly higher and, in aggregate, not equivalent to that in a single ovulation-per-cycle regime. Specifically, on average the level was approximately 2500 pg/hr versus 1500 pg /hr in second versus first/single ovulation respectively. To

put this in perspective, in previous studies baseline corticosterone levels in female *X. laevis* were 1000+/-600 pg/hr and were increased to around 1350 pg/hr by changing the tank background colour from black to white whereas shipping frogs increased corticosterone secretion to around 2500 pg/hr [32].

Corticosterone levels were back to baseline level by the time of the second ovulation in the double-ovulated group (i.e. after seven days) and were very similar between the first two cycles and the last two, despite repeated double-ovulation in the intervening cycles. The fact that there appeared to be no long-lasting effect on corticosterone levels of either ovulation regime is important since chronically raised levels of stress hormones implement a host of detrimental physiological changes in animals, including amphibians such as impaired immune function [35], suppressed reproductive physiology [36], growth impairments [37], and altered cognition [38]. Moreover, although not directly relevant to this study, elevated corticosterone in amphibians disrupts metamorphosis with subsequent negative impacts on development [39]. Although we observed no deleterious effects over seven three-month cycles and no signs of chronic difference between the control and double-ovulated animals, it cannot be completely ruled out that there would be some difference over an even longer period. Since lab *Xenopus laevis* can live for over 20 years, this is a relevant question. There may therefore be room for further modifications to the ovulation protocol, for example increasing the time between first and second ovulation, either for a few more days or even to multiple weeks so that the regime is effectively a single ovulation protocol but with a 1.5-month rather than 3-month rest interval.

Elevated corticosterone levels following ovulation were not surprising in light of the link between the hypothalamic pituitary gonadal axis and hypothalamic pituitary adrenal axis demonstrated in animals, including amphibians [36]. Elevation of corticosterone concentrations post ovulation thus probably reflects the physiological demands of ovulation rather than an abnormally harmful event, especially since estrogens, which rise prior to ovulation, increase glucocorticoid levels in several mammalian and amphibian species and are needed to mobilise energy stored [36, 40, 41]. The higher corticosterone elevation following the second ovulation in the double ovulation group may thus reflect the need to mobilise more physiological resources, rather than more suffering. Importantly however, even the higher corticosterone levels we recorded for the second ovulation were within levels routinely experienced by laboratory *Xenopus laevis*.

Thus, we conclude that although the double-ovulation protocol does make more demands on each frog, these are within the sustainable range for this species and remain within the U.K. harm categorisation of Mild rather than Moderate or Severe.

## Materials and methods

The study was performed according to the UK Home Office regulations and protocols approved by the King's College London Animal Welfare and Ethical Review Board and conducted under the HO Project licence PP5705975 to J.B.A.G. Animals were kept in a Tecniplast Xenoplus recirculating water tank system under established standard temperature, salinity and pH conditions for this species (i.e. 18–20˚C, NaCl added to achieve conductivity 1500 ± 200 mS, pH 7.0–7.8).

### Animal housing, grouping and scheduling

80 adult virgin *Xenopus laevis* (African claw-toed frog) females, all approximately 36 months old from the same batch, were obtained from the European Xenopus Resource Centre, University of Portsmouth) and were tagged with an ID chip (Pet-id Microchips, 8 mm) by injection ventral to the right lateral line, giving each a 15-digit numeric ID. They were then weighed,

assigned weight-matched pairs and then the members of each pair randomly assigned to the single- or double-ovulation group. The animals were housed in 10 groups ("home" tanks) of 8 females (four weight-matched pairs) so that each group included frogs in the two treatment arms in identical shared conditions. Between receiving the frogs and the beginning of the ovulations, two frogs (one frog in each treatment arm) showed signs of infectious disease and were removed from the study. During the trial, a further seven frogs died or were removed from the study due to operational reasons (six drowned due to a tank overflow malfunction and one succumbed to an unidentified systemic infection).

For ovulation, six frogs from a given home tank (all four double-ovulation-arm and two of the single-ovulation arm frogs) were placed in ovulation tanks randomly numbered from 1 to 6 for human chorionic gonadotrophin (hCG) injection (see below). After ovulation, all frogs were returned to the same "home" tank. The following week, six frogs from the same tank (all four double-ovulation-arm again and the other two of the single-ovulation arm frogs) were again placed in randomly numbered ovulation tanks for hCG injection. Frogs were moved by animal care staff according to chip ID number so that they were blind to the treatment arm assignment, while the researchers performing the hCG injections and all subsequent steps were always seeing only groups of six randomly numbered frogs and so were also blind to the treatment arm assignment. Egg counts, fertilisation rates, and corticosterone assays were labelled according to the date and ovulation tank number. Blinding to the group and holding tank assignment schedule was broken only at the end of the study.

After the two-week ovulation period, each frog was rested (i.e., left undisturbed in their home tank other than by routine feeding, cleaning, and veterinary inspection) for a subsequent twelve or thirteen weeks (twelve for the double-ovulated and twelve or thirteen for the single-ovulated at random). Feeding was *ad libitum*, i.e. excess chow was added to the tanks and leftovers removed after 30–40 minutes, so that no change in protocol was needed to deal with post-ovulation recovery feeding.

The procedure was followed for seven ovulation cycles in total (21 months between September 2020 and July 2022 inclusive). Data for first two cycles (acute effects) and last two cycles (chronic effects) are reported.

## Induction of ovulation

The females were injected in the dorsal lymph sac with 500U of hCG (i.e. 250 μl of hCG resuspended in 5 ml of distilled water; 10,000 IU, CG10, Sigma Aldrich) 16 hours before the experimental procedure to induce ovulation. The induced frogs were maintained in ovulation tanks, numbered 1 to 6, containing 5L of system water each. We chose not use the "squeeze" method for expressing eggs from the females but instead left them to lay their eggs in a solution that maintains fertilisability. Typically, this method produces a lower levels of fertilization than can be achieved with the "squeeze" method, but was used to minimise handling stress and maximise consistency (the priority for a side-by-side comparision). Thus, on the ovulation day, the frogs were transferred in 2L of 1X egg laying solution each (ELS; 8.8 mM NaCl, 1.6 mM KCl, 12mM Tris-base, 0.4 mM Na2HPO$_4$, 1.6 mM NaHCO$_3$, 0.8 mM MgSO$_4$.7H2O, ph7.6 with glacial acetic acid) for 60 to 90 minutes to promote egg laying. The eggs were harvested and processed for *in vitro* fertilization. The frogs were then transferred in a diluted ELS solution (2L 1X ELS and 3L system water) for 6 hours 30 minutes before being transferred back into 5L of system water.

## In vitro fertilisation

The eggs were harvested in 1X ELS. The salts contained in the 1X ELS solution were diluted in distilled water and then the eggs were rinsed once in 0.1X Modified Barth's Saline (MBS: 8

mM NaCl, 0.1 mM KCl, 0.07 mM CaCl$_2$, 0.1 mM MgSO$_4$, 0.5 mM HEPES, 0.25 mM NaHCO$_3$, pH7.8). The excess of 0.1X MBS solution was removed from around the eggs. An *in vitro* fertilization was performed with a concentrated suspension of sperm, extracted from fresh male testes (The European Xenopus Resource Centre, University of Portsmouth), mixed with distilled water. To allow the sperm entry, the eggs were incubated for 5 minutes at room temperature before flooding the embryos with 0.1X MBS. The successfully fertilised eggs presented a rotation of their animal pole about 20 minutes post-fertilization and divided after about 1 hour 30 minutes at room temperature. Room temperature was defined as 20˚C unless otherwise specified.

## Determination of the egg number

Eggs were counted from photographs of the eggs laid into the tank over the ovulation day (approximately 8 hours following the 16 hours overnight following hCG injection). The frog was removed from the tank, the eggs allowed to settle, and the overlying medium/water poured off before the tank was placed on a light box to silhouette the eggs and a photograph taken with a mobile phone camera. This generally gave images that allowed semi-automated segmentation using Fiji (ImageJ) [42]. Images were transformed into an 8-bit then binarized with the Auto Local Threshold function, (selecting the Sauvola method with a radius of 15). Watershed and Fill holes functions were then applied. The median filter despeckle was then used to reduce the image noise. The built-in Analyze Particles function was used, choosing a minimum particle diameter of 80 pixels to exclude debris. Sometimes egg numbers were very low ($< 500$) or, alternatively, eggs were hard to segment because of clumping or debris (e.g. food particles stuck to the jelly), in which case eggs were counted manually using the Fiji multi-point tool.

## Determination of fertilisation rate

After 1.5–2 hours of egg laying into ELS, a sample of up to 300 eggs was collected from each tank into Petri dishes and fertilised following ELS removal using testis macerated in a minimum volume of 0.1 X MBS. "Fertilisation" was defined as the percentage of eggs dividing to the 64-128-cell stage (i.e., fertilised successfully as demonstrated by cleavage).

## Corticosterone assays

Corticosterone excretion was measured for each frog before hCG injection (i.e. before ovulation, giving a "baseline") and on the morning after egg laying, i.e. for one set of six frogs on a Monday and a Wednesday and a second set of six frogs on a Wednesday and Friday with eggs being laid, counted, collected, and fertilised on Tuesday and Thursday.

Corticosterone excretion was measured as previously described [32]. In brief, females were placed in small tanks (footprint 19 x 12 cm) in 1 L of system water (see details above) for 1 hour. The water was collected for each female and vacuum-filtered using glass filtration apparatus (Cole-Parmer Labglass BP-1755-000 Filtration Assembly, 47mm dia, 300 mL) and a Capex 8C 230V STD, X37-950 vacuum pump, (Charles Austin Pumps) to remove debris (regurgitated food particles, fecal matter, shed skin, etc.). Filtration was performed three times with successively finer filters (paper coffee filters (Aeropress), Whatman Number 1 (Cat. No.1001-090) pore size 11 μm, and Protran nitrocellulose (Amersham, cat. No.10600020) pore size 0.45 μm). Water samples were then pumped at 25 ml/min using a peristaltic pump (Ecoline, Ismatec) through activated solid-phase extraction cartridges (Sep-pak ® Plus C18, Waters Ltd., U.K.) previously primed with 5 ml HPLC-grade 100% methanol and 5 ml distilled water. Cartridges were then washed with 5 ml distilled water and stored at -20˚C until elution. Samples from six frogs were processed in parallel and time from water collection to freeze-down was typically 3 hours.

Corticosterone bound to the hydrophobic silica components of the Sep-Pak cartridges was eluted as described in [32] into borosilicate glass tubes (16 mm x 100 mm, Fisherbrand) using 4 ml of ethyl acetate injected through the tube at a steady rate over 2 minutes. The ethyl acetate was evaporated under nitrogen at 37˚C and samples were resuspended in 500 μl of ELISA buffer (0.039 M NaH$_2$PO4, 0.06 M Na$_2$HPO$_4$, 0.15 M NaCl, 1% (w/v) BSA, pH 7.00) and vortexed for 20 mins at 1600 rpm (Multi-Reax, Heidolph). Samples were stored at -20˚C until analysis. Corticosterone in samples was quantified by enzyme-linked immunosorbent assay (ELISA) as described in [32]. In brief, 50 μl of capture antibody CJM006 (rabbit anti- corticosterone-3-CMO-BSA polyclonal antibody, produced by C. Munro, University of California, Davis, CA) at a 1 in 20,000 dilution in 0.05 M sodium carbonate buffer (Na$_2$CO$_3$,/NaHCO$_3$, pH 9.6), was used to coat plates at 4˚C overnight. Plates were washed three times with ELISA wash buffer (i.e., phosphate buffered saline plus Tween 20, pH 7.4 (137 mM NaCl, 2.7 mM KCl, 1.8 mM KH2PO4, 10 mM Na2HPO4, 0.05% Tween 20, pH 7.4) 50 μl of ELISA buffer was added to all wells, followed by 50 μl of corticosterone standard (C2505 Sigma-Aldrich) in duplicate (30,000 pg/ml– 58.6 pg/ml) or duplicates of 50 μl sample diluted, 1:2 in ELISA buffer. 50 μl of corticosterone-hydrogen peroxidase conjugate (C. Munro, University of California, Davis, CA) at a 1 in 40,000 dilution in ELISA buffer was added to all wells and plates incubated, with gentle shaking, 100 rpm, at room temperature in the dark for 3 hours. Plates were washed as before and 100 μl per well of ABTS substrate (0.04 mM 2,2'-azino-di-(3-ethyl-benzthiazoline sulfonic acid) diammonium salt (ABTS), 1.6 mM H$_2$O$_2$, 0.05 M citrate pH 4.0) was added. Plates were incubated at room temperature with shaking at 600 rpm for approximately 1 hour until the OD$_{405nM}$ reached a value of 1. Samples were re-assayed if the coefficient of variation (CV) was > 5%.

Quality controls were run in duplicate on each plate and comprised a low (1:4 dilution) and high (1:2 dilution) concentration aliquot from a pool of samples from adult female *Xenopus laevis* which were part of a previous study [32]. Inter-assay low and high CVs for the whole study were obtained by averaging the separate inter-assay CVs from each of the 4 cycles (N = 47 plates). An intra-assay CV for the completed study was similarly computed from the mean intra-assay CV from each cycle which was in turn computed from the average CVs for each of the low and high QCs run on the 47 plates. All samples from a given collection date (i.e., containing anonymized samples from both treatment arms) were assayed together (N = 21 different assay days) to minimize effects of assay variation. High and low inter-assay CVs for the two to three plates run on a single day provided a mean 'intra-day' plate CV reflecting assay variation within groups of matched samples run on different plates. Inter-assay CVs for low and high concentration QCs were 7.2% and 8.7% respectively (N = 47 plates). Intra-assay CVs for low and high concentration QCs were 2.9% and 4.1% respectively. Mean intra-day plate CVs for low and high QCs were 3.6% and 3.7%. Sensitivity was determined as the lowest concentration of corticosterone in the working range of the assay and measured as 234 pg/ml.

## Statistical methods

Descriptive statistics, graphs, normality, and initial hypothesis tests (stated in the text) were generated using GraphPad Prism. Due to known changes in the housing/maintenance conditions of the frogs during the nearly two years of the trial, including feed changes in cycles 3, 4 and 5, as well as multiple personnel changes, we decided that we could not treat the cycles as a reliable time-series and so one-way rather than two-way ANOVA was used for multiple testing. Further hypothesis testing was carried out using the R statistics package [43]. Code is freely available from the authors upon request.

None of the datasets had Normal distributions but log-values did (Shapiro-Wilk test) and so hypothesis tests used log values for ANOVA and Tukey post-hoc tests.

All data generated are included in Supporting Information S1 Appendix.

## Supporting information

**S1 Appendix. Primary data for figures (egg counts, fertilisation rates, corticosterone assays, body weights).**
(XLSX)

## Acknowledgments

The authors would like to thank Michelle Scutter and the rest of the KCL New Hunt's House BSU staff for their diligent care of the animals and for significant work in putting out the right individuals for ovulation according to the schedule to facilitate the trial blinding.

## Author Contributions

**Conceptualization:** Charlotte A. Hosie, Tessa E. Smith, Jeremy B. A. Green.

**Data curation:** Barbara Vacca, Jo Arnold, Jeremy B. A. Green.

**Formal analysis:** Chloe Moss, Michael A. R. Green, Jeremy B. A. Green.

**Funding acquisition:** Charlotte A. Hosie, Tessa E. Smith, Jeremy B. A. Green.

**Investigation:** Chloe Moss, Barbara Vacca, Jo Arnold, Chantal Hubens, Dominic M. Lynch, James Pegge, Jeremy B. A. Green.

**Methodology:** Barbara Vacca, Jo Arnold, Charlotte A. Hosie, Tessa E. Smith, Jeremy B. A. Green.

**Project administration:** Tessa E. Smith, Jeremy B. A. Green.

**Supervision:** Charlotte A. Hosie, Tessa E. Smith, Jeremy B. A. Green.

**Validation:** Chloe Moss, Michael A. R. Green.

**Visualization:** Chloe Moss.

**Writing – original draft:** Barbara Vacca, Tessa E. Smith, Jeremy B. A. Green.

**Writing – review & editing:** Chloe Moss, Barbara Vacca, Jo Arnold, Chantal Hubens, Dominic M. Lynch, James Pegge, Michael A. R. Green, Charlotte A. Hosie, Tessa E. Smith, Jeremy B. A. Green.

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
