## [Decision Letter · Decision Letter 0]

11 Mar 2024

PONE-D-24-05117A double ovulation protocol for Xenopus laevis produces doubled fertilisation yield and moderately transiently elevated corticosterone levels without loss of egg qualityPLOS ONE

Dear Dr. Green,

Thank you for submitting your manuscript to PLOS ONE. After careful consideration, we feel that it has merit but does not fully meet PLOS ONE’s publication criteria as it currently stands. Therefore, we invite you to submit a revised version of the manuscript that addresses the points raised during the review process.

The two reviewers have raised a number of concerns over some details that need explaining in the text.  Besides, modifications in the arrangement of figures (merging two specific figures, etc.) have been suggested.

We look forward to receiving your revised manuscript.

Kind regards,

Jyotshna Kanungo, Ph.D.

Academic Editor

PLOS ONE

“This work was funded by the UK National Centre for the Replacement Refinement and Reduction of Animals in Research (NC3Rs) grant NC/S000933/1-GREEN to J.B.A.G. and T.E.S.”

Additional Editor Comments:

The two reviewers have raised concerns over some details that need explaining in the text. Besides, modifications in the arrangement of figures (merging two specific figures, etc.) have been suggested.

Reviewers' comments:

Reviewer's Responses to Questions

**Comments to the Author**

1. Is the manuscript technically sound, and do the data support the conclusions?

Reviewer #1: Yes

Reviewer #2: Yes

2. Has the statistical analysis been performed appropriately and rigorously? 

Reviewer #1: Yes

Reviewer #2: Yes

3. Have the authors made all data underlying the findings in their manuscript fully available?

Reviewer #1: Yes

Reviewer #2: Yes

4. Is the manuscript presented in an intelligible fashion and written in standard English?

Reviewer #1: Yes

Reviewer #2: Yes

5. Review Comments to the Author

Reviewer #1: This article describes the use of a double ovulation procedure for Xenopus as opposed to the traditional single ovulation procedure. Overall the results are sound and show that a second quicker induced ovulation can work with Xenopus. However there are a few questions raised by the study that need to be addressed.

1) The corticosterone levels are raised by the second ovulation, similar to those in shipped frogs. Is this something a lab would want to put the frogs through on a regular basis? Since frogs can live for 20+ years what are the long term effects?

2) No mention of age of frogs or whether they were virgins. This should be addressed.

3) Seven frogs were removed from the trial and perished. Why? This is not common for Xenopus laevis.

4) Was feeding increased between ovulations?

5) Does amount of hormone change egg output? Would the use of PMSG before hCG change?

6) Seems that the rate of in vitro fertilizations was low across the board. Why did the researchers not simply squeeze the eggs out as most labs do instead of collecting in 1X MBS? Normally with X. laevis labs achieve near 100% fertlization.

Reviewer #2: Moss and colleagues propose a double ovulation protocol for Xenopus leavis as opposed to the single ovulation protocol currently used as standard frog fertilisation protocol. Traditionally the laboratory protocol for obtaining Xenopus eggs is to induce one round of ovulation and egg laying, and then to wait 3 months before anther induction. The rationale behind the double ovulation approach is that in the wild, frogs undergo successive matings within the mating seasons. Therefore, it may be expected that two rounds in rapid succession of ovulation prior to the prolong interval would be possible without harming the animals, and would double egg yield. The authors state that the egg laying capacity of Xenopus leavis is potentially higher than observed during the standard procedure, and a double ovulation would result in a higher fertilisation yield, a reduction in animal use in accordance with the 3Rs (Replacement, Reduction and Refinement) and benefits in cost-savings. The authors thoroughly test egg yield, egg quality, and stress hormones in the mother from this approach and find that the double ovulation approach indeed increases effective egg yield with acceptable perturbation to the mother. With minor changes to the text and figures, the protocol described by Moss and colleagues represents a valuable contribution to the scientific community using Xenopus leavis as a model organism.

Minor points:

The authors compared a double ovulation protocol to the single ovulation standard protocol over a time period of 21 months and evaluated the egg number, fertilisation rate, and stress response (corticosterone levels) of the animals during the ovulation cycles of the study. However, there is no reference given for the standard protocol or a reference to a published protocol on which the author's standard protocol is based (line 65). In the standard ovulation protocol described in this paper, female frogs are injected with chorionic gonadotropin, thereby stimulating ovulation followed by egg laying. After ovulation, frogs rest for a 3-month period before the procedure is repeated. In the proposed double ovulation protocol, female frogs are re-injected seven days after the first ovulation, and the 3-month resting period starts after the second ovulation. This procedure was repeated for seven 3-month cycles in total, and data was presented for ovulation cycles 1, 2, 6 and 7. Although it is sufficient to show data from cycles 1, 2 and 6,7 it is not clear why data from cycles 3, 4 and 5 are not shown. It would have been interesting to see the extent of variability throughout the trial.

The arrangements of the plots, consistency in labelling and references to the figures in the text and legends of figures 1–3 need to be checked carefully and corrected accordingly. The plots in the figures lack labelling, such as A,B, C, etc. However, in the text, references to Fig.x A,B are given. For more details, see “additional points” below.

Data in Figures 1, 2 and 3 are shown in two different representations within the figure. On one hand, with bar plots, and on the other hand, with box plots. It is not clear from the text why this representation was chosen. For simplicity and a better understanding of the data, I recommend skipping the bar plots and using box plots only. Moreover, I suggest combining plots from Figure 1 and Figure 2 into one figure. In this way, plots representing egg count can be directly compared to the fertilisation rate.

In the discussion, the authors strongly focus on the limitations of the study. I recommend to slightly shift the focus to the findings and data rather than the variabilites or missed opportunities in experimental measurements (lines 244 -247). It is absolutely valid to point out the limitations of the study, but it is important to put these limitations into context with the presented data and point out the positive aspects of the findings. This has been done for most of the paragraphs of the discussion but is missing in paragraphs 244-247. I would suggest re-formulating this paragraph. In line 244, it says, “Second, some further measurements that could have been informative were not made.” The question “Why?” immediately comes to mind. Give arguments, why the shown data is still representative and enough to support the study.

Overall, the data presented in this study supports the advantages of the double ovulation protocol over the single ovulation protocol proposed by the authors. Despite variabilites introduced during the trial, such as changes in animal care staff, feeding changes, and technical issues in the animal facility, the authors applied appropriate statistical testing and could show that the double ovulation protocol resulted in a higher egg yield without loss of egg quality compared to the standard single ovulation protocol. Although elevated levels of stress were experienced by the frogs during the second ovulation protocol, these levels were only transient and fall within an acceptable range for Xenopus leavis.

Additional points:

• Figure 1 and Figure 2: I would suggest merging Figure 1 and Figure 2 and showing only the box plots. I recommend labelling the plots in the figure properly for better reference in the text and figure legends (e.g., A,B). Figure legends need to be updated accordingly.

• Figure 1, right panel, top plot: Here, the x-axis is labelled “ovulation group”. The other plots have the x-axis labelled “frog ovulation group”. I strongly suggest labelling the x-axis consistently across plots.

• Figure 2 and Figure 3, the two top plots have their x-axis labelled “ovulation group” and the bottom plots with “frog ovulation group”. Again, I strongly suggest labelling plots consistently.

• Figure 3: The y-axis of the plots in figure 3 is labelled “Corticosterone level” but no unit is given, such as [pg/hr].

• Line 59: The citation given here is a bit confusing, as it is not consistent with the citation format within the rest of the manuscript, and it is not clear if, where, and when the survey has been published.

• Line 142: Here, a reference is given to Figure 2 “Fig.2A”. However, no “A” is assigned to a plot in the figure.

• Line 182, there is a reference to “Fig. 3A,B”. However, there is no “A” or “B” assigned to the plots presented in figure 3.

• Line 227: The sentence ends with an open exclamation mark. Is the sentence in the figure legend complete?

• Line 286: There is no information given on the age of the frogs. Were the freshly purchased frogs the same age? Was the age well-balanced between the different groups?

6. PLOS authors have the option to publish the peer review history of their article (what does this mean?). If published, this will include your full peer review and any attached files.

Reviewer #1: No

Reviewer #2: No

---

## [Author Response · Author response to Decision Letter 0]

30 Mar 2024

Responsese are detailed in the Response to Reviewers. As requested, we have explained additional details in the text and merged two of the figures.

---

## [Decision Letter · Decision Letter 1]

10 May 2024

PONE-D-24-05117R1A double ovulation protocol for Xenopus laevis produces doubled fertilisation yield and moderately transiently elevated corticosterone levels without loss of egg qualityPLOS ONE

Dear Dr. Green,

Thank you for submitting your manuscript to PLOS ONE. After careful consideration, we feel that it has merit but does not fully meet PLOS ONE’s publication criteria as it currently stands. Therefore, we invite you to submit a revised version of the manuscript that addresses the points raised during the review process.

The reviewers have pointed out a few discrepancies in the text and figure labeling.  The authors are advised to address these issues.  Once these minor corrections are made to the manuscript, it will only be reviewed by the Academic Editor without undergoing further external review. 

We look forward to receiving your revised manuscript.

Kind regards,

Jyotshna Kanungo, Ph.D.

Academic Editor

PLOS ONE

Journal Requirements:

Additional Editor Comments:

A reviewer has pointed out a few discrepancies in the text and figure labeling. The authors are advised to address these issues. Once these minor corrections are made to the manuscript, it will only be reviewed by the Academic Editor without undergoing further external review.

Reviewers' comments:

Reviewer's Responses to Questions

**Comments to the Author**

1. If the authors have adequately addressed your comments raised in a previous round of review and you feel that this manuscript is now acceptable for publication, you may indicate that here to bypass the “Comments to the Author” section, enter your conflict of interest statement in the “Confidential to Editor” section, and submit your "Accept" recommendation.

Reviewer #1: All comments have been addressed

Reviewer #2: All comments have been addressed

2. Is the manuscript technically sound, and do the data support the conclusions?

Reviewer #1: Yes

Reviewer #2: Yes

3. Has the statistical analysis been performed appropriately and rigorously? 

Reviewer #1: Yes

Reviewer #2: Yes

4. Have the authors made all data underlying the findings in their manuscript fully available?

Reviewer #1: Yes

Reviewer #2: Yes

5. Is the manuscript presented in an intelligible fashion and written in standard English?

Reviewer #1: Yes

Reviewer #2: Yes

6. Review Comments to the Author

Reviewer #1: The authors have addressed all the comments although i do not agree with their response to point #6. They stated that they did not squeeze the frogs to lay eggs because they wanted to maximise consistency of eggs for fertilization, but then in same sentence later they state that they get lower fertilization this way.

Reviewer #2: Moss et al. have adequately addressed and implemented the suggested comments. Besides a few comments on text formatting, the manuscript is acceptable for publication.

- In the revised manuscript showing tracked changes, a different title is given, however the revised manuscript without tracked changes has the original title.

- Figure 1 C/D: y-axis label in C shows “Fertilization Rate %”, y-axis label in D shows “Fertilization Rate (%)”. Add the parentheses in C.

- Figure 1 G: It looks like the y-axis label has a bigger font size than the other axis labels in this figure. Adjust font size for consistency.

- 205-206: In the legend of Figure 2 the reference to A, B, C, D is bold. However, in all other figure legends letters are not bold. Make them consistent.

- 227-230: Here, the parenthesis at the end of the sentence is open “…significant difference between the groups (two-tailed paired t-test…”.

- 262: “It would indeed be interesting to track differences the recovery time for corticosterone levels after ovulation, but for this study…”. Check the sentence – it looks like a word is missing between “differences” and “the recovery time”.

7. PLOS authors have the option to publish the peer review history of their article (what does this mean?). If published, this will include your full peer review and any attached files.

Reviewer #1: No

Reviewer #2: No

---

## [Author Response · Author response to Decision Letter 1]

10 Jun 2024

All reviewers' comments have been addressed.

---

## [Editor Report · Decision Letter 2]

13 Jun 2024

A double ovulation protocol for Xenopus laevis produces doubled fertilisation yield and moderately transiently elevated corticosterone levels without loss of egg quality

PONE-D-24-05117R2

Dear Dr. Green,

We’re pleased to inform you that your manuscript has been judged scientifically suitable for publication and will be formally accepted for publication once it meets all outstanding technical requirements.

Kind regards,

Jyotshna Kanungo, Ph.D.

Academic Editor

PLOS ONE

Additional Editor Comments (optional):

The authors have addressed all the minor text changes that were warranted.
---

## [Editor Report · Acceptance letter]

10 Jul 2024

PONE-D-24-05117R2 

PLOS ONE

Dear Dr. Green, 

I'm pleased to inform you that your manuscript has been deemed suitable for publication in PLOS ONE. Congratulations! Your manuscript is now being handed over to our production team.

Kind regards, 

on behalf of

Dr. Jyotshna Kanungo 

Academic Editor

PLOS ONE